# MSPE: Multi-Scale Patch Embedding Prompts Vision Transformers to Any Resolution

**Wenzhuo Liu**[1,2], **Fei Zhu**[3], **Shijie Ma**[1,2], **Cheng-Lin Liu**[1,2]*
[1]School of Artificial Intelligence, UCAS
[2]State Key Laboratory of Multimodal Artificial Intelligence Systems, CASIA
[3]Centre for Artificial Intelligence and Robotics, HKISI-CAS
{liuwenzhuo2020, zhufei2018, mashijie2021}@ia.ac.cn, liucl@nlpr.ia.ac.cn

## Abstract

Although Vision Transformers (ViTs) have recently advanced computer vision tasks significantly, an important real-world problem was overlooked: adapting to variable input resolutions. Typically, images are resized to a fixed resolution, such as 224x224, for efficiency during training and inference. However, uniform input size conflicts with real-world scenarios where images naturally vary in resolution. Modifying the preset resolution of a model may severely degrade the performance. In this work, we propose to enhance the model adaptability to resolution variation by optimizing the patch embedding. The proposed method, called Multi-Scale Patch Embedding (MSPE), substitutes the standard patch embedding with multiple variable-sized patch kernels and selects the best parameters for different resolutions, eliminating the need to resize the original image. Our method does not require high-cost training or modifications to other parts, making it easy to apply to most ViT models. Experiments in image classification, segmentation, and detection tasks demonstrate the effectiveness of MSPE, yielding superior performance on low-resolution inputs and performing comparably on high-resolution inputs with existing methods. Code is available at https://github.com/SmallPigPeppa/MSPE.

## 1 Introduction

Vision Transformers (ViTs) [1, 2, 3, 4] have achieved significant success in various computer vision tasks, becoming a viable alternative to traditional convolutional neural networks [5, 6, 7, 8]. ViT divides an image into multiple patches, converts these patches into tokens via the patch embedding layer, and feeds them into the Transformer model [9, 10]. The token representations are usually obtained using a convolutional neural network (CNN). Early CNN architectures like AlexNet [11] were designed for fixed-size images (e.g., 224x224). For easy comparison, this setting has been maintained by subsequent image recognition models, including ViT [2]. For fitting neural network input layer size, ViTs typically resize the input image to a fixed resolution [2] and divide it into a specific number of patches [12, 13, 14]. This practice restricts ViT models to processing single-resolution inputs. However, fixed input sizes conflict with real-world scenarios where image resolution varies due to camera devices/parameters, object size, and distance. This discrepancy can significantly degrade ViT's performance on different-resolution images. Changing the preset size requires retraining, and every resolution necessitates an individual model. Therefore, a natural question arises: ***Is it possible for a single ViT model to process different resolutions directly?***

---

*Corresponding author.

[2]The term *resolution* refers to the width and height of images input into neural networks. In this paper, *resolution* and *image size* are used interchangeably.

38th Conference on Neural Information Processing Systems (NeurIPS 2024).

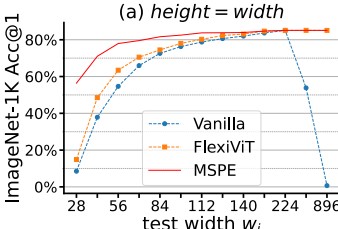 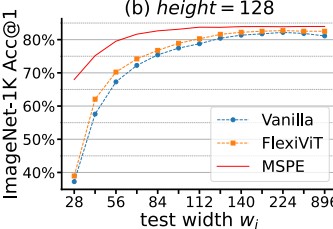

Figure 1: **MSPE results on ImageNet-1K.** We loaded a ViT-B model pre-trained on ImageNet-21K from [19] and evaluated: (a) Height equals width, ranging from 28×28 to 896×896, and (b) Fixed height=128, width ranging from 28 to 896. Vanilla ViT performance drops with size/aspect ratio changes; FlexiViT [15] significantly improves performance, and our method surpasses FlexiViT.

A few recent methods have been proposed for this problem: FlexiViT [15] uses a novel resizing method allowing flexible patch size and token length in ViT. ResFormer [16] enhances resolution adaptability through multi-resolution training and a global-local positional embedding strategy. NaViT [17] leverages example packing [18] to adjust token lengths during training, boosting efficiency and performance. However, these methods have two limitations. **First**, some of them are incompatible with existing transformer models, preventing the effective use of pre-trained ViTs, and thus lead to high re-training costs when applied. For example, models like NaViT and ResFormer require training the entire model to achieve multi-resolution performance. **Second**, their multi-resolution performance is insufficient. For example, FlexiViT demonstrates only slight performance improvement when the input images are smaller than the preset size.

We intuitively believe that an ideal solution should be compatible with most existing ViT models and perform well at any size and aspect ratio. To this end, we propose a method called **M**ulti-**S**cale **P**atch **E**mbedding (MSPE). It replaces the patch embedding layer in standard ViT models without altering the other parts of the model. This design makes MSPE compatible with most ViT models and allows for low-cost, direct application. Specifically, our method uses a set of learnable adaptive convolution kernels instead of fixed ones, which can automatically adjust the size and aspect ratio of the kernels based on the input resolution. It directly converts images into patch embeddings without modifying the input size or aspect ratio. The results in Figure 1 show that this simple method significantly improves performance, for example, increasing ImageNet-1K accuracy up to 47.9% across various resolutions. Our contributions are as follows:

- By analyzing resolution adaptability in ViT models, we identify the patch embedding layer as the crucial component and provide a low-cost solution.

- We propose Multi-Scale Patch Embedding (MSPE), which enhances ViT models by substituting the standard patch embedding layer with learnable, adaptive convolution kernels, enabling ViTs to be applied on any input resolution.

- Experiments demonstrate that with minimal training (only five epochs), MSPE significantly enhances performance across different resolutions on classification, segmentation, and detection tasks.

## 2 Preliminaries and Analysis

### 2.1 Vision Transformer Models

Vision Transformers (ViTs) leverage the capabilities of the transformer model [20, 21], initially developed for natural language processing, to address vision tasks [22, 23, 24]. ViTs primarily consist of the patch embedding layer and Transformer encoder.

**Patch Embedding** converts an image $x \in \mathbb{R}^{h \times w \times c}$ in the input space $\mathcal{X}$ into a sequence of tokens $\{z_i\}_{i=1}^N$, where $z_i \in \mathbb{R}^d$ is a vector in patchification space $\mathcal{Z}$. This transformation is achieved via patch embedding layer $g_\theta : \mathcal{X} \to \mathcal{Z}$, parameterized by $\theta$. Specifically, the function $g_\theta$ is implemented as convolution $\text{conv}_\theta$, using kernel $w_\theta \in \mathbb{R}^{h_k \times w_k \times d}$ and bias $b_\theta \in \mathbb{R}^d$. The kernel size is $(h_k, w_k)$ and the stride is $(h_s, w_s)$. In existing ViTs, patch embedding methods are categorized into non-overlapping and overlapping types:

- **Non-overlapping patch embedding:** In standard Vision Transformers (e.g., ViT [2]), the stride of $\text{conv}_{\boldsymbol{\theta}}$ matches the kernel size, *i.e.*, $h_s = h_k$ and $w_s = w_k$. The number of tokens $N$ is calculated by $\left\lfloor \frac{h}{h_k} \right\rfloor \times \left\lfloor \frac{w}{w_k} \right\rfloor$.
- **Overlapping patch embedding:** In models like PVT [4] and MViT [25], the stride is smaller than the kernel size, *i.e.*, $h_s < h_k$, $w_s < w_k$. The number of tokens $N$ is calculated by $\left\lceil \frac{h-h_k}{h_s} + p \right\rceil \times \left\lceil \frac{w-w_k}{w_s} + p \right\rceil$, where $p$ is the padding size.

**Transformer Encoder** adds position encodings $\text{pos}_i$ to the token sequence, modifying it to $\{\boldsymbol{z}_i'\}_{i=1}^N = \{\boldsymbol{z}_i + \text{pos}_i\}_{i=1}^N$. A class token $\boldsymbol{z}_{\text{cls}}$ for global semantics is added to the front, *i.e.*, $\{\boldsymbol{z}_{\text{cls}}, \boldsymbol{z}_1', \ldots, \boldsymbol{z}_N'\}$. This sequence feeds into the Transformer Encoder $\text{Enc}_{\boldsymbol{\phi}}(\boldsymbol{z})$, parameterized by $\boldsymbol{\phi}$. For classification, only the class token $\boldsymbol{z}_{\text{cls}}$ is used to produce a probability distribution, and other tokens $\{\boldsymbol{z}_1', ..., \boldsymbol{z}_N'\}$ are discarded.

## 2.2 Problem Formulation

In real-world scenarios, images have variable resolutions due to variations in camera devices, parameters, object scale, and imaging distance. Our task is to ensure the model adapts to different resolutions. To this end, we resize the same image to multiple resolutions and optimize the model for these diverse inputs. Here is the formal definition of this task:

Firstly, the process of resizing (e.g. bilinear reize) is formally defined as a linear transformation:

$$\text{resize}_r^{r*}(\boldsymbol{o}) = B_r^{r*}\text{vec}(\boldsymbol{o}). \tag{1}$$

where $\boldsymbol{o} \in \mathbb{R}^{h \times w}$ is any input, the resolution $r$ of $(h, w)$ transforms to $r*$ of $(h*, w*)$ after resizing, using the transformation matrix $B_r^{r*} \in \mathbb{R}^{h*w* \times hw}$. Each channel of $\boldsymbol{x}$ is resized independently.

For input $(\boldsymbol{x}, y)$ in dataset $\mathcal{D}$, the learning objective is to minimize the loss function $\ell$ (e.g., cross-entropy loss) across a series of resolutions $\{r_i\}_{i=1}^M$, optimizing performance at each resolution from $r_1$ to $r_M$:

$$\min_{\boldsymbol{\theta}, \boldsymbol{\phi}} \mathbb{E}_{(\boldsymbol{x}, y) \sim \mathcal{D}}[\ell(\text{Enc}_{\widehat{\phi}}(g_{\widehat{\theta}}(B_r^{r_i}\boldsymbol{x})), y] \quad \forall r_i \in \{r_i\}_{i=1}^M$$

$$\text{s.t. } \mathbb{E}_{(\boldsymbol{x}, y) \sim \mathcal{D}}[\ell(\text{Enc}_{\widehat{\phi}}(g_{\widehat{\theta}}(\boldsymbol{x})), y)] \leqslant \mathbb{E}_{(\boldsymbol{x}, y) \sim \mathcal{D}}[\ell(\text{Enc}_{\boldsymbol{\phi}}(g_{\boldsymbol{\theta}}(\boldsymbol{x})), y)] + \epsilon, \epsilon \geqslant 0, \tag{2}$$

where $Enc_{\widehat{\phi}}(\boldsymbol{z})$ and $g_{\widehat{\theta}}(\boldsymbol{x})$ are Transformer encoder and patch embedding layer, respectively. The slack variable $\epsilon$ allows minor loss increments in the well-trained model $\text{Enc}_{\boldsymbol{\phi}}(g_{\boldsymbol{\theta}}(\boldsymbol{x}))$.

The central challenge is maintaining acceptable performance across different resolutions from $r_1$ to $r_M$ but keeping the performance of original resolution $r$, which means adjustments of $\theta$ and $\phi$ must be careful. In this work, we confirm the key role of patch embedding and only optimizing $g_{\boldsymbol{\theta}}$:

$$\widehat{\boldsymbol{\theta}} \in \arg\min_{\widehat{\boldsymbol{\theta}}} \mathbb{E}_{(\boldsymbol{x}, y) \sim \mathcal{D}}[\ell(\text{Enc}_{\boldsymbol{\phi}}(g_{\widehat{\theta}}(B_r^{r_i}\boldsymbol{x})), y] \quad \forall r_i \in \{r_i\}_{i=1}^M. \tag{3}$$

## 2.3 Pseudo-inverse Resize

To solve the optimization problems stated in Eq. (2) and (3), an intuitive solution appears to ensure that embedding layer $g_{\boldsymbol{\theta}}$ produces consistent features on different resolutions. For this purpose, FlexiViT [15] proposed PI-resize, which adjusts the embedding kernel to ensure uniform outputs:

$$\{\widehat{\boldsymbol{\omega_\theta}}, \widehat{\boldsymbol{b_\theta}}\} = \widehat{\boldsymbol{\theta}} \in \arg\min_{\widehat{\boldsymbol{\theta}}} \mathbb{E}_{\boldsymbol{x} \sim \mathcal{X}}[(g_{\widehat{\theta}}(B_r^{r_i}\boldsymbol{x}) - g_{\boldsymbol{\theta}}(\boldsymbol{x}))^2] \quad \forall r_i \in \{r_i\}_{i=1}^M,$$

$$\widehat{\boldsymbol{\omega_\theta}} \in \arg\min_{\widehat{\boldsymbol{\omega_\theta}}} \mathbb{E}_{\boldsymbol{x} \sim \mathcal{X}}[(\langle \boldsymbol{x}, \boldsymbol{\omega_\theta} \rangle - \langle B_r^{r_i}\boldsymbol{x}, \widehat{\boldsymbol{\omega_\theta}} \rangle)^2]. \tag{4}$$

When upscaling with $r_i > r$, the analytic solution for Eq. (4) is $\widehat{\boldsymbol{\omega_\theta}} = B_r^{r_i}(B_r^{r_i T}B_r^{r_i})^{-1}\boldsymbol{\omega_\theta}$, denote as $(B_r^{r_i T})^+\boldsymbol{\omega_\theta}$:

$$\langle B_r^{r_i}\boldsymbol{x}, \widehat{\boldsymbol{\omega_\theta}} \rangle = \boldsymbol{x}^T B_r^{r_i T}B_r^{r_i}(B_r^{r_i T}B_r^{r_i})^{-1}\boldsymbol{\omega_\theta} = \boldsymbol{x}^T\boldsymbol{\omega_\theta} = \langle \boldsymbol{x}, \boldsymbol{\omega_\theta} \rangle. \tag{5}$$

When downscaling with $r_i < r$, the matrix $B_r^{r_i T}B_r^{r_i}$ is non-invertible. Under the assumption $\mathcal{X} = \mathcal{N}(0, 1)$, it is proven that $(B_r^{r_i T})^+\boldsymbol{\omega_\theta}$ is the optimal solution. In summary, Pseudo-inverse resize (PI-resize) is defined as follows:

$$\text{PI-resize}_r^{r*}(\boldsymbol{w}) = (B_r^{r_i T})^+\text{vec}(\boldsymbol{w}). \tag{6}$$

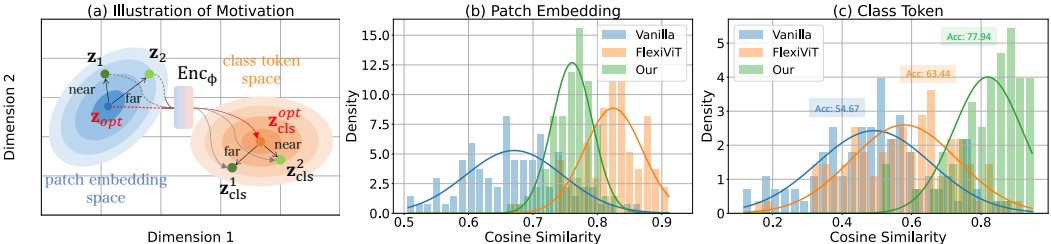

Figure 2: Similarity in patch embeddings does not guarantee optimal performance (a). We confirm this by evaluating the accuracy and cosine similarity of: (b) patch embeddings $\{z_i\}_{i=1}^{N}$ from 56x56 and 224x224 images, and (c) class tokens $z_{\text{cls}}$ from 56x56 and 224x224 images.

## 2.4 Motivation

However, is this optimization target in FlexiViT [15] truly appropriate? We suggest that there are two problems with Equation 4: **First**, it is a stricter sufficient condition of Eq. (3), but it ignores the impact of the encoder $\text{Enc}_{\phi}(z)$ and the objective function $\ell$. **Second**, at lower resolutions, *i.e.* $r_i < r$, this goal has no analytical solution; PI-resize is just an approximation assuming $x \sim \mathcal{N}(0,1)$, resulting in significant performance degradation.

Moreover, we intuitively suspect that similarity in patch embeddings does not ensure the best performance. As Figure 2(a) illustrates, features derived from patch embeddings are transformed into the classification feature space, namely the class token space, through the Transformer encoder $\text{Enc}_{\phi}$. The gradient directions of these two feature spaces may not align, resulting in image features that are close to optimal in patch embedding being far from the optimal class token after encoder processing.

A more effective method is directly adjusting the weights $w_{\theta}$ of embedding layer using the objective function $\ell$ (Eq. (3)). To verify this idea and our assumptions, we evaluate a well-trained ViT-B/16 model [2] (pre-trained on ImageNet-21K, 224x224, 85.10% accuracy). We measure the cosine similarity between patch embeddings $\{z_i\}_{i=1}^{N}$ and the class token $z_{cls}$ at resolutions of 56x56 and 224x224. The results in Figures 2 (b) and (c) show that FlexiViT has higher patch embedding similarity and classification accuracy compared to the vanilla model; however, our method significantly outperforms FlexiViT with even lower patch embedding similarity. These results confirm that our analysis is reasonable.

## 3 Method

As discussed in Section 2.4, optimizing patch embedding layers through objective function is simple but effective. Motivated by this, we propose **M**ulti-**S**cale **P**atch **E**mbedding (MSPE). It divides the resolution domain into different ranges $\{r_i\}_{i=1}^{M}$, each using a shared-weight patch embedding layer that adjusts size through PI-resize. MSPE can replace the patch embedding layers in most ViT models, including overlapping and non-overlapping types. Our method is illustrated in Figure 3 and presented below.

### 3.1 Architecture of MSPE

MSPE only changes the patch embedding layer of the ViT model, making it directly applicable to a well-trained ViT model. As demonstrated in Figure 3, we introduce the following architectural modifications.

**Multiple patching kernels.** The typical patch embedding layer $g_{\theta}$ employs a single convolution size with parameters $(\omega_{\theta}, b_{\theta})$, making it unsuitable for varying image resolutions. To overcome this, MSPE incorporates $K$ convolutions with differenet kernel sizes $\{g_{\theta}^1, \ldots, g_{\theta}^K\}$, where each $g_{\theta}^i$ is parameterized by $(w_{\theta}^i, b_{\theta}^i)$, to support a broader range of input image sizes.

**Adaptive patching kernels.** Although $K$ convolutions of different kernel sizes improve adaptability, they cannot cover all possible resolutions. Setting a unique size convolution kernel for each resolution is unrealistic. In MSPE, the size and ratio of the kernel $(w_{\theta}^i, b_{\theta}^i)$ are adjustable rather than fixed. Specifically, for input $x$ of any resolution $(h, w)$, the corresponding kernel size $(h_k, w_k)$ is

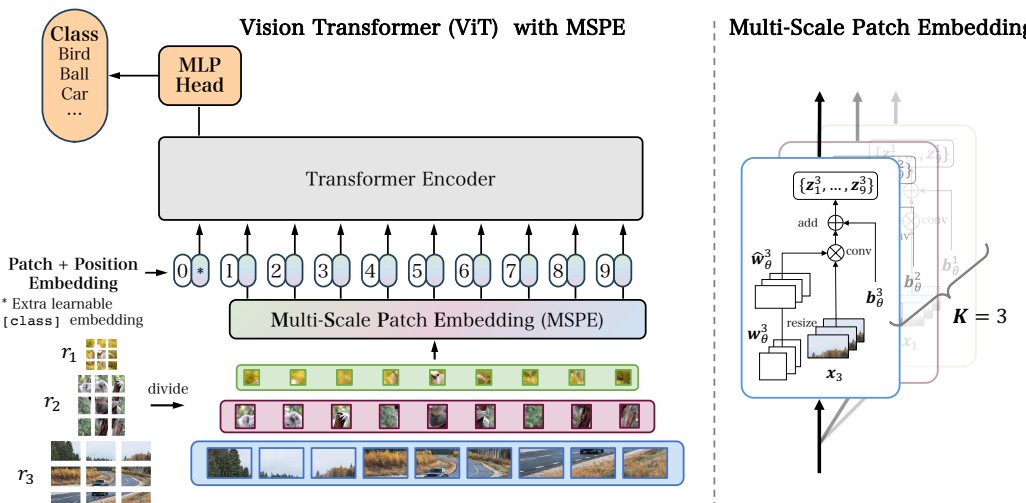

Figure 3: Illustration of the ViT model [2, 3] with MSPE. MSPE only replaces the patch embedding layer in the vanilla model, making well-trained ViT models to be directly applied to any size and aspect ratio. In our method, the patch embedding layer has several variable-sized kernels. The Transformer encoder is shared and frozen.

$(\lfloor h/N \rfloor, \lfloor w/N \rfloor)$. Using Eq. (6), we adjust $\boldsymbol{w}_\theta^i$ to the corresponding size $\widehat{\boldsymbol{w}}_\theta^i \in \mathbb{R}^{h_k \times w_k}$. As shown in Figure 3 (b), it can be directly convolved with the image patch.

Another module requiring careful design is the position embedding, which must correspond to the feature map size after patch embedding. In ViT [2] and DeiT [3], the position embedding is bilinearly interpolated to match the feature map size, a method known as *resample absolute pos embed*, and it has proven effective during fine-tuning [3]. In MSPE, we discover that bilinear interpolation of position embedding satisfies our requirements. To simplify the architecture and minimize changes to the vanilla model, we follow this dynamic positional embedding method.

## 3.2 Learning Objectives

MSPE optimizes the patch embedding layer $g_\theta$ through the objective function but does not explicitly constrain the embedded tokens $\{\boldsymbol{z}_i\}_{i=1}^N$ across different resolutions. Specifically, MSPE is based on multi-resolution training (mixed resolution training), similar to methods like ResFormer and NaViT. The training process is as follows.

Firstly, the mixed resolution $\{r_i\}_{i=1}^M$ is divided into $K$ subsets $\{S_k\}_{k=1}^K$, i.e. $\bigcup_{k=1}^K S_k = \{r_i\}_{i=1}^M$, and $r_k$ is randomly sampled from $S_k$. The patching kernel weights $\boldsymbol{\theta}_k$ are shared within $S_k$ and transformed into the corresponding weights $\widehat{\boldsymbol{\theta}_k}$ for each $r_i$ according to Eq. (6), The loss function is defined as:

$$\mathcal{L}_{\boldsymbol{\theta}}(\boldsymbol{x}, y) = \sum_{i=1}^K \ell[(\mathrm{Enc}_\phi(g_{\widehat{\boldsymbol{\theta}_i}}(B_r^{r_i}\boldsymbol{x})), y] + \lambda \cdot \ell[(\mathrm{Enc}_\phi(g_{\boldsymbol{\theta}}(\boldsymbol{x})), y], \tag{7}$$

where $\ell$ is the task loss function (e.g., cross-entropy loss), and $\lambda$ is a hyperparameter to prevent performance degradation. We optimize only the patch embedding parameters $\boldsymbol{\theta}$ during training, setting the learning rate of $\phi$ to zero. Algorithm 1 in Appendix E.2 details the training procedure of MSPE and PyTorch-style implementation.

## 3.3 Inference on Any Resolution

For an input $\boldsymbol{x}$ with any resolution $r^*$, previous models resize the image to a fixed resolution $r$. The inference procedure is:

$$\hat{y} = \mathrm{Enc}_\phi(g_{\boldsymbol{\theta}}(B_{r^*}^r \boldsymbol{x})), \tag{8}$$

With MSPE, the model can infer directly from the original image without resizing or altering its aspect ratio. The process is:

$$\hat{y} = \mathrm{Enc}_\phi(g_{\boldsymbol{\theta}^*}(\boldsymbol{x})), \tag{9}$$

Table 1: ImageNet-1K Top-1 accuracy across 28×28 to 448×448 resolutions: Our method was only trained for 5 epochs, while ResFormer [16] was trained for 200 epochs, all methods based on the same well-trained model.

| | Method | Resolution | | | | | | | | | | | |
|---|---|---|---|---|---|---|---|---|---|---|---|---|---|
| | | 28 | 42 | 56 | 70 | 84 | 98 | 112 | 126 | 140 | 168 | 224 | 448 |
| *ViT* | Vanilla [2] | 8.52 | 37.86 | 54.67 | 65.94 | 72.58 | 76.29 | 78.75 | 80.62 | 82.00 | 83.66 | 85.10 | 53.81 |
| | ResFormer [16] | 2.21 | 19.29 | 45.41 | 59.92 | 69.30 | 74.34 | 77.92 | 77.48 | 79.61 | 81.30 | 83.04 | 81.06 |
| | FlexiViT [15] | 14.86 | 48.52 | 63.44 | 70.53 | 74.47 | 78.03 | 80.24 | 82.28 | 83.10 | 84.70 | 85.10 | 85.11 |
| | MSPE | **56.41** | **71.02** | **77.94** | **79.54** | **81.63** | **82.51** | **83.75** | **83.81** | **83.94** | **84.70** | **85.10** | **85.11** |
| *DeiTIII* | Vanilla [30] | 0.14 | 2.45 | 17.93 | 40.30 | 57.23 | 67.01 | 72.80 | 76.20 | 79.62 | 83.51 | 85.66 | 62.29 |
| | ResFormer [16] | 1.08 | 20.27 | 46.52 | 60.78 | 70.28 | 75.28 | 78.80 | 78.79 | 80.48 | 82.13 | 83.47 | 82.53 |
| | FlexiViT [15] | 3.76 | 26.44 | 49.99 | 62.72 | 70.47 | 74.55 | 77.57 | 79.22 | 80.69 | 83.51 | 85.66 | 85.53 |
| | MSPE | **48.71** | **65.66** | **75.37** | **77.78** | **80.76** | **81.68** | **83.49** | **83.28** | **83.75** | **84.86** | **85.66** | **85.53** |
| *PVT* | Vanilla [4] | 8.78 | 28.20 | 47.52 | 53.54 | 61.11 | 68.64 | 76.45 | 78.29 | 80.21 | 81.73 | 83.12 | 73.48 |
| | FlexiViT [15] | 16.64 | 48.13 | 60.21 | 65.39 | 70.44 | 73.27 | 76.70 | 78.29 | 80.21 | 81.73 | 83.12 | 81.57 |
| | MSPE | **34.20** | **60.76** | **72.09** | **75.39** | **77.39** | **78.08** | **80.36** | **81.12** | **81.59** | **82.00** | **83.12** | **82.43** |

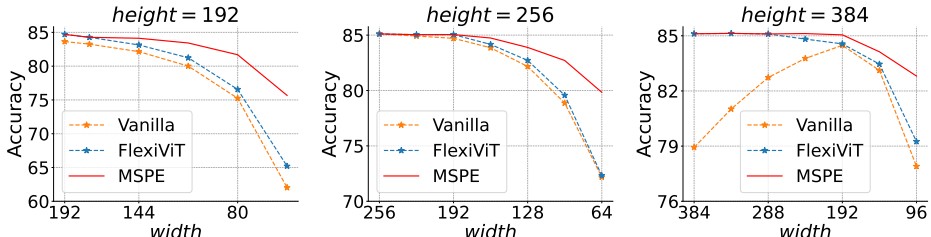

Figure 4: ImageNet-1K Top-1 accuracy curves, fixed heights at 192, 256, and 384. Results show MSPE directly applied across varying input ratios and enhancing performance.

where $\theta^*$ is the patching weight that matches the resolution $r^*$. Details on the computation of $\theta^*$ are provided in Appendix E.1 and Eq.(10).

## 4 Experiments

**Datasets.** We conduct experiments on 4 benchmark datasets: ImageNet-1K [26] for classification tasks, ADE20K [27] and Cityscapes [28] for semantic segmentation, and COCO2017 [29] for object detection.

**Backbone networks.** Our method is assessed on three ViT models, including ViT-B [2] and DeiT-B III [30] (non-overlapping patches); PVT v2-B3 [31] (overlapping patches). ViT and DeiT III are pre-trained using the ImageNet-21K and ImageNet-22K datasets.

**Implementation details.** MSPE is trained using SGD optimizer for ***five epochs***, with a learning rate of 0.001, momentum of 0.9, weight decay of 0.0005, and batch size of 64 per GPU. To validate our model, we implement ViT [2] and other networks [30, 4, 25] via open-sourced `timm` library for classification, ViTDet [32] via MMDetection [33] for object detection, and SETR [34] via MMSegmentation [35] for segmentation; additional details are available in Appendix B.

### 4.1 Image Classification

**Comparison with FlexiViT.** As shown in Table 1 and Figure 4, ViT models using non-overlapping (like ViT and DeiT III) and overlapping patch embeddings (like PVTv2 and MViTv2) significantly lose accuracy when the input resolution varies. This is consistent with the results from [15, 16, 17], demonstrating that the vanilla ViT models are not adaptable to input resolution changes. This issue primarily arises from their patch embedding layers failing to adjust to varying resolutions. This leads to high-level features shifting after the patch tokens are fed into the Transformer encoder and significantly degrading performance. FlexiViT shows remarkably stable performance at upscaled resolutions (e.g., 448x448) by ensuring consistency of patch tokens across different resolutions, outperforming vanilla models. However, as Section 3 analyzes, FlexiViT still struggles with downscaling.

Table 2: Comparative results of semantic segmentation on ADE20K and Cityscapes, using well-trained SETR Naive [34] as the segmentation model (ViT-L backbone), evaluated by mIOU, mACC, and F1-score.

| | Method | 128x128 | | | 256x256 | | | 512x512 | | |
| --- | --- | --- | --- | --- | --- | --- | --- | --- | --- | --- |
| | | mIOU | mACC | F1-score | mIOU | mACC | F1-score | mIOU | mACC | F1-score |
| ADE20K | Vanilla | 2.89 | 4.91 | 4.64 | 7.48 | 10.81 | 15.88 | 45.39 | 55.70 | 59.71 |
| | FlexiViT | 31.44 | 38.86 | 44.13 | 42.97 | 52.48 | 57.21 | 45.39 | 55.70 | 59.71 |
| | MSPE | **39.65** | **49.01** | **51.37** | **44.39** | **53.73** | **58.90** | 45.39 | 55.70 | 59.71 |

| | Method | 192x192 | | | 384x384 | | | 768x768 | | |
| --- | --- | --- | --- | --- | --- | --- | --- | --- | --- | --- |
| | | mIOU | mACC | F1-score | mIOU | mACC | F1-score | mIOU | mACC | F1-score |
| Cityscapes | Vanilla | 13.33 | 19.49 | 24.62 | 20.10 | 27.34 | 26.84 | 77.58 | 84.94 | 86.72 |
| | FlexiViT | 59.78 | 68.65 | 72.74 | 74.60 | 82.81 | 84.58 | 77.58 | 84.94 | 86.72 |
| | MSPE | **65.90** | **74.84** | **77.21** | **75.79** | **83.40** | **85.35** | 77.58 | 84.94 | 86.72 |

Table 3: Comparative results of object detection and instance segmentation on COCO2017, employing well-trained ViTDeT [32] as the detection model (ViT-B backbone), pre-trained on ImageNet-1K via MAE [36].

| | Method | $AP^b$ | $AP^b_{50}$ | $AP^b_{75}$ | $AP^b_s$ | $AP^b_m$ | $AP^b_l$ | $AP^m$ | $AP^m_{50}$ | $AP^m_{75}$ | $AP^m_s$ | $AP^m_m$ | $AP^m_l$ |
| --- | --- | --- | --- | --- | --- | --- | --- | --- | --- | --- | --- | --- | --- |
| 1024 | Vanilla | 0.52 | 0.72 | 0.57 | 0.35 | 0.56 | 0.66 | 0.46 | 0.69 | 0.50 | 0.27 | 0.49 | 0.64 |
| 512 | Vanilla | 0.34 | 0.50 | 0.36 | 0.19 | 0.37 | 0.45 | 0.29 | 0.47 | 0.31 | 0.14 | 0.31 | 0.44 |
| | FlexiViT | 0.44 | 0.63 | 0.47 | 0.27 | 0.47 | 0.57 | 0.39 | 0.61 | 0.41 | 0.21 | 0.42 | 0.56 |
| | MSPE | **0.47** | **0.68** | **0.49** | **0.29** | **0.50** | **0.62** | **0.42** | **0.64** | **0.43** | **0.22** | **0.44** | **0.61** |
| 256 | Vanilla | 0.03 | 0.05 | 0.03 | 0.01 | 0.04 | 0.05 | 0.03 | 0.05 | 0.03 | 0.01 | 0.04 | 0.05 |
| | FlexiViT | 0.19 | 0.31 | 0.20 | 0.10 | 0.23 | 0.29 | 0.17 | 0.29 | 0.17 | 0.07 | 0.19 | 0.28 |
| | MSPE | **0.30** | **0.42** | **0.29** | **0.16** | **0.34** | **0.44** | **0.27** | **0.39** | **0.27** | **0.12** | **0.30** | **0.43** |

Our method significantly boosts accuracy with targeted optimization goals, outperforming FlexiViT across various resolutions and aspect ratios.

**Comparison with ResFormer and NaViT.** ResFormer improves performance by multi-resolution training across 128x128, 160x160, and 224x224 resolutions. However, the modified ViT architecture does not suit networks like PVT and MViT that use overlap patch embedding. Moreover, ResFormer trains for 200 epochs, but our method requires only five epochs. NaViT keeps the original aspect ratio and trains with mixed resolutions from 64x64 to 512x512, leveraging a larger JFT pre-training dataset and longer training cycles (up to 920,000 steps). Table 1 and Figure 5 show that these state-of-the-art methods improve the vanilla model remarkably. Compared to ResFormer and NaViT, MSPE achieves superior multi-resolution performance with far fewer training resources, which proves the essential role of optimized patch embedding layers.

## 4.2 Semantic Segmentation

To validate the effectiveness of MSPE in semantic segmentation, we test the SETR [34] model on ADE20K [27] and Cityscapes [28] datasets, with the vanilla model trained at 512x512 and 768x768 resolutions, respectively. Results in Table 2 show that FlexiViT significantly enhances the vanilla model's performance in semantic segmentation (e.g., 59.78 vs. 13.33 on the mIOU metric). This confirms that adjusting the patch embedding layer is effective for pixel-level tasks, demonstrating its critical role in enhancing multi-resolution robustness. Moreover, MSPE consistently outperforms FlexiViT across various resolutions, proving our method is ready for pixel-dense tasks in real-world scenarios.

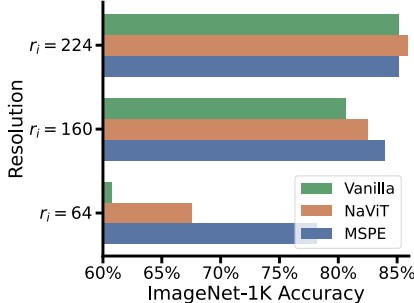

Figure 5: Comparison of MSPE, Vanilla, and NaViT: only NaViT was pre-trained on the JFT dataset, baseline results come from [17].

## 4.3 Object Detection

In our experiments on the COCO2017 dataset for object detection and instance segmentation, we utilize the ViTDeT [32] model with ViT-B (pre-trained on ImageNet-1K via MAE [36]) and

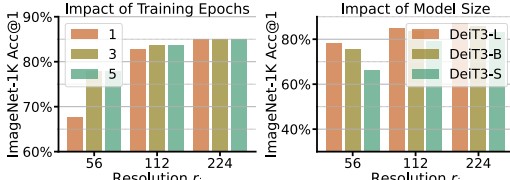

Figure 6: Comparison results: (a) different training epochs; (b) model sizes of S, B, and L.

Figure 7: Comparison results: (a) hyperparameter $\lambda$; (b) differenet kernel count $K$.

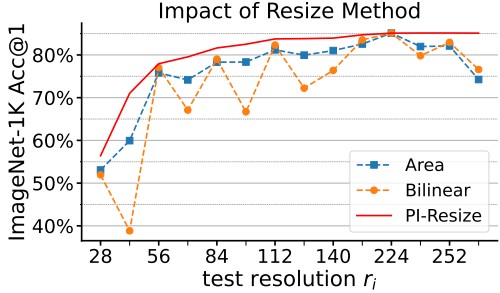

Figure 8: Comparison results of different resizing methods in MSPE. PI-resize shows the best performance and robustness.

Table 4: Parameter and computational cost of the patch embedding layer $g_{\theta}$ in MSPE and Vanilla models, the parameter count of the entire model remains nearly unchanged.

| Model | Method | Parms($g_{\theta}$) | FLOPs($g_{\theta}$) | Parms(all) |
|---|---|---|---|---|
| ViT-B | Vanilla | **590.59K** | 0.92 | **86.57M** |
| | MSPE | 1108.99K | **0.01~0.92** | 87.09M |
| PVTv2-B3 | Vanilla | **10.50K** | 2.83 | **45.25M** |
| | MSPE | 29.70K | **0.13~2.83** | 45.27M |

employed Mask R-CNN [37] as the detection head. During the evaluation, we replace the ViT's patch embedding layer with MSPE or FlexiViT, keeping the rest of the architecture unchanged ( aligned with classification and segmentation ). As shown in Table 3, MSPE significantly improves the multi-resolution performance of well-trained detection models. This result is consistent with those of classification and segmentation tasks, demonstrating the effectiveness of our method across different visual tasks.

## 4.4 Ablation Study and Analysis

**Training epochs.** Figure 6 (a) presents the performance of MSPE on different training epochs $(1, 3, 5)$. It is observed that MSPE significantly enhances performance with only a few training epochs. The models trained for 3 and 5 epochs show similar performance, with no significant improvement from additional epochs. Thus, we train MSPE for 5 epochs in our experiments.

**Model size.** For evaluating the impact of model size on MSPE, we test on ImageNet-1K across different sizes of DeiT III models (Small (S), Base (B), Large (L)), all pre-trained on ImageNet-22K. As shown in Figure 6 (b), the results of the larger model DeiT-L and the smaller model DeiT-S align with the main experiment; the larger model yields higher accuracy at different resolutions. This result demonstrates the effectiveness of our method across models of different sizes.

**Hyperparameters.** We conduct ablation studies of hyperparameters $\lambda$ in Figure 7 (a). When lambda is set to 0, the learning of patch embedding is too flexible, leading to inadequate alignment with the original parameter space. Lambda values of 1 and 2 lead to similar performances; this hyperparameter is set to 1 in our experiments.

**Kernel count $K$.** Figure 7 (b) shows the impact of different kernel quantities on model performance. In MSPE, the resolution $r_i$ is divided into $K$ subsets, with each subset sharing the patch embedding layer weights. Therefore, the number of subsets $K$ equals the number of patchify kernels in MSPE. The results indicate that slightly increasing $K$ can improve performance. Specifically, when $K = 3$ and $K = 4$, the model performance is nearly identical, and additional kernels provide little improvement. This suggests that patch embedding parameters can be shared across different resolutions. In our method, $K$ is set to 4.

**Resizing method.** In our method, the patch embedding weights are dynamically resized for images with different sizes and ratios, denoted as adaptive kernels in Section 3.1. To evaluate the effect of resizing methods on MSPE, we load an ImageNet-21K pre-trained ViT-B model from [19] and train MSPE using different resizing methods, such as standard linear resizing. As shown in Figure 8, the

Table 5: Comparison results of image resizing and MSPE on ImageNet-1K Top-1 accuracy.

| | Method | Resolution | | | | | | | | | | | |
|---|---|---|---|---|---|---|---|---|---|---|---|---|---|
| | | 28 | 42 | 56 | 70 | 84 | 98 | 112 | 126 | 140 | 168 | 224 | 448 |
| *ViT* | IMG-resize | 52.99 | 67.51 | 74.06 | 76.86 | 79.16 | 80.25 | 80.88 | 82.04 | 82.68 | 83.99 | 85.10 | 84.92 |
| | MSPE | **56.41** | **71.02** | **77.94** | **79.54** | **81.63** | **82.51** | **83.75** | **83.81** | **83.94** | **84.70** | **85.10** | **85.11** |
| *DeiT* | IMG-resize | 39.97 | 58.40 | 66.60 | 71.16 | 74.20 | 76.37 | 78.15 | 79.67 | 80.39 | 82.01 | 83.39 | 82.46 |
| | MSPE | **45.01** | **62.54** | **71.92** | **73.72** | **77.09** | **78.18** | **80.29** | **79.94** | **80.58** | **82.14** | **83.39** | **83.39** |
| *PVT* | IMG-resize | 32.09 | 52.12 | 61.54 | 67.36 | 71.31 | 74.05 | 76.32 | 78.14 | 79.45 | 80.95 | 83.12 | 82.34 |
| | MSPE | **34.20** | **60.76** | **72.09** | **75.39** | **77.39** | **78.08** | **80.36** | **81.12** | **81.59** | **82.14** | **83.12** | **83.08** |

results indicate that PI-resize consistently outperforms other common resizing methods, aligning with findings from the FlexiViT [15].

**Parameters and computation overhead.** As shown in Table 4, we analyze the impact of MSPE on the parameter and computational cost of the vanilla VIT model. MSPE modifies only the patch embedding layer $g_\theta$, increasing its parameter count by 2x~3x. However, MSPE provides a more flexible approach to calculating patch embeddings, significantly reducing computational costs compared to the original method. The total parameter count of the model remains nearly unchanged because the patch embedding layer is an extremely small component.

**Image resizing v.s. MSPE.** Table 5 shows the comparison results of image resizing (IMG-resize) and MSPE on ImageNet-1K. IMG-resize adjusts images of different resolutions to a preset resolution (e.g., 224x224) during testing, whereas our method keeps the original image size and aspect ratio unchanged. In real-world scenarios, resizing small-size images to a larger size incurs digitization noise and increased computation costs, and model performance is not optimal at different resolutions. The results indicate that our method comprehensively outperforms IMG-resize, suggesting that model inference on the original images is a viable option.

# 5   Related Work

Most relevant to our work is FlexiViT [15], which proposed to resize the patch embedding weights, enabling flexible patch sizes and token lengths in ViT models. Pix2struct [38] supports variable aspect ratios by a novel positional embedding method, enhancing efficiency and performance in chart and document understanding. ResFormer [16] enables resolution adaptability through multi-resolution training and a global-local positional embedding strategy. NaViT [17] enhances efficiency and performance by example packing [18] to adjust token lengths for different resolutions during training. Compared to these methods, MSPE only changes the patch embedding layer and achieves better performance. This justifies the patch embedding layer's key role in adapting ViT models for different resolutions.

In CNN-based models, Mind the Pooling [39] addresses overfitting on resolution by introducing SBPooling to replace max-pooling, enabling CNNs to process different resolutions. Learn to Resize [40] uses a learnable resizing layer to replace bilinear interpolation. Another study [41] examined the relationship between training and testing resolutions, showing that training at slightly lower resolutions than testing can improve performance. Networks like Resolution Adaptive Networks (RANet) [42] and Dynamic Resolution Networks (DRNet) [43] use multiple sub-models to choose the appropriate resolution and model based on task difficulty, thus enhancing model efficiency and resolution adaptability.

# 6   Conclusion

To make ViT models compatible with images of different sizes and aspect ratios, we propose MSPE to replace the traditional patch embedding layer for accommodating variable image resolutions. MSPE uses multiple variable-sized patch kernels and selects the best parameters for different resolutions, eliminating the need to resize the original image. Extensive experiments demonstrate that MSPE performs well in various visual tasks (image classification, segmentation, and detection). Particularly, MSPE yields superior performance on low-resolution inputs and performs comparably on high-resolution inputs with previous methods. Our method has the potential for application in various vision tasks, and can be extended by optimizing the embedding layer and transformer encoder jointly.

# 7 Acknowledgments and Disclosure of Funding

This work has been supported by the National Natural Science Foundation of China (NSFC) grant U20A20223 and the InnoHK program.

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

# A Limitation

Our paper does not consider the impact of position embedding on model performance. This is because existing work [2, 3] has shown that linear interpolation of position embedding can achieve acceptable performance. Our experimental results validate the feasibility of this strategy. Moreover, we aim to utilize our method within most existing ViT models, hence we do not alter the position embedding method. In the future, our method can be extended with the positional encoding strategies proposed in related works, such as ResFormer [16] and Pix2struct [38].

# B Training details

## B.1 Multi-resolution training

Multi-resolution training has become a widely used paradigm [44, 45], randomly altering crop size and aspect ratio during training. This training strategy is critical in multi-resolution vision models. In FlexiViT [15] training, the size of image patches and the kernel size of the patch embedding layer are randomly changed from 8x8 to 48x48. In ResFormer [16], input images are resized to $128 \times 128$, $160 \times 160$ and $224 \times 224$ and simultaneously training. In NaViT [17], images retain their original size and aspect ratio and are packed into a single pack for training, denoted as mixed-resolution training. The range of its training resolutions is 64 to 256. Our method is most similar to FlexiViT, randomly resizing each image batch to different resolutions (from 56 to 256) during training.

## B.2 Data augmentation

**Classification.** Our method leverages `timm` implementations of ViT [2], DeiT III [30], and PVT v2 [31] models for classification tasks. The training data augmentations include RandomResizedCrop, RandomHorizontalFlip, and ColorJitter. For Resfomer [16], we follow the official-released code. Its training data augmentation includes Auto-Augment [46], RandAugment [47], random erasing [48], MixUp [49], and CutMix [50].

**Semantic segmentation.** We use the SETR [34] model implemented by MMSegmentation [35]. We adhere data settings in MMSegmentation, including RandomResizedCrop, RandomFlip, and Photo-MetricDistortion. PhotoMetricDistortion involves a series of transformations: random brightness, random contrast, random saturation, random hue, and converting color between HSV and BGR.

**Object detection.** We employ the ViTDeT [32] model within the MMDetection [33] for object detection and instance segmentation tasks. We utilize the configurations from MMDetection, which include RandomResizedCrop and RandomFlip.

# C Compute resources

This paper conducts experiments on a machine equipped with two AMD EPYC 7543 32-core processors; each slotted with 32 cores supporting two threads per core. The machine has 496 GB of memory and 8* NVIDIA GeForce RTX 4090 graphics cards. Our method significantly reduces computational resources compared to previous approaches, and all main experiments on ImageNet-1K are completed within 3 hours. In contrast, methods like ResFormer [16] require over 50 hours of training on this machine.

# D More experimental results

## D.1 Impact of aspect ratio

In Section 4.1, we fixed the height and varied the width, ensuring the height was greater than the width. We also tested the accuracy when fixing the width and varying the height, in which case the width was greater than the height. It can be seen in Figure 9 that MSPE still achieved significant performance improvements.

Furthermore, comparing the accuracy of resolutions $(h, w)$ and $(w, h)$ when $h > w$, we observe that the accuracy of $(h, w)$ resolution was higher than $(w, h)$ resolution. This aligns with the findings

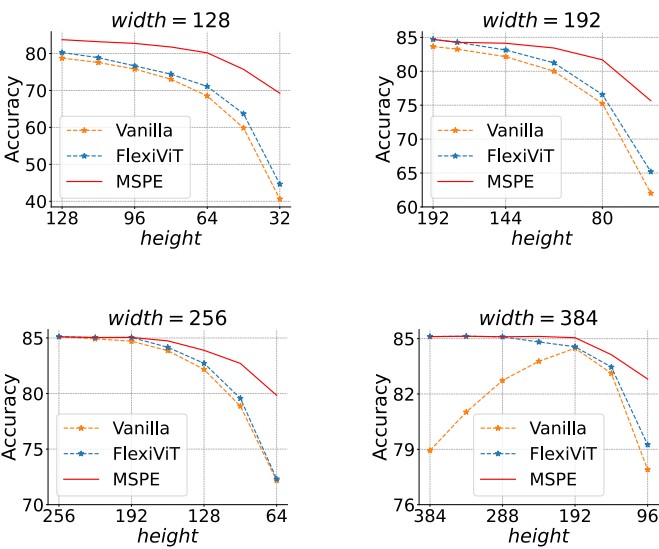

Figure 9: ImageNet-1K Top-1 accuracy curves, fixed width at 128, 192, 256, and 384.

of NaViT [17], which indicate that a larger proportion of images in ImageNet-1K is $h > w$. This suggests that maintaining the original aspect ratio during inference needs to be considered.

## D.2 Results on Higher Resolution

Below are experimental results for high-resolution images. We resize ImageNet images (processed to 224x224) up to 4096x4096 and evaluate models with non-overlapping patch embedding (ViT) and overlapping patch embedding (PVT). The results in Table 6 show that MSPE can be effectively applied to high-resolution images, as the shape and size of the patch embedding kernels dynamically adjust based on input resolution and have multiple convolution kernels corresponding to different resolution ranges.

Table 6: Comparison results of higher resolution.

| | Method | Resolution | | | | | | | | | |
|---|---|---|---|---|---|---|---|---|---|---|---|
| | | 224 | 448 | 672 | 896 | 1120 | 1792 | 2240 | 2688 | 3360 | 4032 |
| *ViT* | Vanilla | 85.10 | 53.04 | 5.17 | 0.68 | 0.33 | 0.13 | 0.08 | 0.08 | 0.08 | 0.08 |
| | MSPE | **85.10** | **85.11** | **85.13** | **85.14** | **85.14** | **85.15** | **85.14** | **85.14** | **85.13** | **85.16** |
| *PVT* | Vanilla | 83.12 | 73.64 | 71.18 | 67.04 | 67.04 | 64.19 | 63.73 | 63.50 | 64.44 | 63.26 |
| | MSPE | **83.12** | **82.44** | **83.11** | **82.76** | **83.02** | **82.89** | **82.93** | **82.65** | **82.97** | **82.58** |

## D.3 Extra training epochs

In our main experiment, MSPE is trained for 5 epochs. We increase the number of epochs to analyze its performance under extended training. As shown in Table 7, the performance improvement with longer training epochs is minimal. This is because MSPE requires learning a very small number of parameters, which can reach optimal performance within just a few epochs of training. This indicates that our method achieves better performance and requires less training overhead, making it highly compatible with well-trained models.

Table 7: Comparison results of extra training epochs.

| Epochs | Resolution | | | | | | | | | | | |
|---|---|---|---|---|---|---|---|---|---|---|---|---|
| | 28 | 42 | 56 | 70 | 84 | 98 | 112 | 126 | 140 | 168 | 224 | 448 |
| 1 | 28.07 | 55.79 | 67.61 | 73.27 | 78.32 | 80.52 | 82.92 | 83.21 | 83.81 | 84.73 | 85.10 | 85.11 |
| 3 | 54.81 | 70.68 | 77.69 | 79.17 | 81.49 | 82.36 | 83.73 | 83.65 | 83.92 | 84.72 | 85.10 | 85.11 |
| 5 | 56.42 | 71.01 | 77.96 | 79.54 | 81.64 | 82.51 | 83.74 | 83.80 | 83.95 | 84.70 | 85.10 | 85.11 |
| 10 | 57.82 | 71.42 | 78.12 | 79.91 | 81.68 | 82.62 | 83.72 | 83.83 | 83.99 | 84.70 | 85.10 | 85.11 |
| 20 | 58.88 | 71.90 | 78.22 | 80.07 | 81.81 | 82.79 | 83.79 | 83.83 | 84.02 | 84.70 | 85.10 | 85.11 |

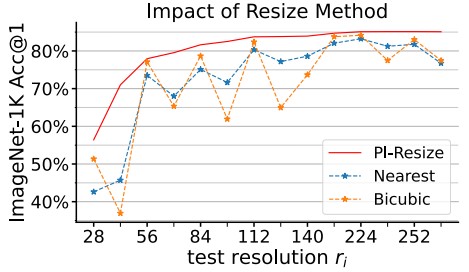
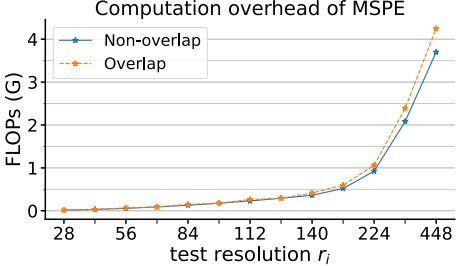

Figure 10: Comparison results of different resizing methods in MSPE.

Figure 11: Computational Overhead of MSPE at Different Resolutions.

### D.4 Other resizing methods

We test other resizing methods in MSPE, including nearest and bicubic[3]. As shown in Figure 10, these traditional resizing methods are not suitable for adjusting the size of neural network parameters, leading to less robust patch embeddings in MSPE training. The experimental results are consistent with Section 4.4, confirming that the combination of PI-resize and MSPE have significant potential applied to multi-resolution models.

### D.5 Computation overhead

We evaluate MSPE's computational overhead across resolutions from 32x32 to 448x448. MSPE dynamically adjusts its parameters based on image resolution. For overlapping patch embedding models like PVT and MViT, MSPE adjusts the convolutional kernel size and stride. For non-overlapping patch embedding models like ViT, MSPE adjusts the convolutional kernel size. As shown in Figure 11, MSPE's computational overhead changes dynamically, avoiding the unnecessary computational costs associated with resizing images to a fixed resolution.

## E  More details on MSPE

In this section, we provide more details about the implementation of MSPE, including modifications to the model structure and the training process, and its application for inference at any resolution.

### E.1 Inference on any resolution

For an input image $\boldsymbol{x}$ with any resolution $r$, we calculate its patch embedding parameters $\boldsymbol{\theta}^*$, composed of $\boldsymbol{w}^*$ and $b^*$. We directly compute the patch embedding $\boldsymbol{z}$ on the original image $\boldsymbol{x}$ as follows: $\boldsymbol{z} = \text{conv}(\boldsymbol{x}, \boldsymbol{w}^*) + b$. After that, embedding $\boldsymbol{z}$ is fed into the Transformer encoder $\text{Enc}(\boldsymbol{z})$ as usual. In this process, calculating $\boldsymbol{\theta}^*$ is a crucial step. We compute $\boldsymbol{\theta}^*$ by finding the resolution $r_i$ in the resolution sequence $r_i{}_{i=1}^M$ that is closest to the original resolution $r^*$ and then resizing its kernel to match the original resolution. This process is formalized as follows:

$$i \in \arg\min_{i=1}^{M} ||r_i - r^*|| \quad \forall r_i \in \{r_i\}_{i=1}^M,$$
$$\boldsymbol{w}^* = (B_{r_i}^{r^*}{}^T)^+ \text{vec}(\boldsymbol{w}_\theta^i) \tag{10}$$
$$\boldsymbol{\theta}^* = \{\boldsymbol{w}^*, b_\theta^i\}.$$

### E.2 MSPE structure and training process

As depicted in Section 3.1, MSPE modifies two parts of the ViT model, as shown in the `class ViT` of Algorithm 1. 1) It adds $K$ patch embedding kernels. 2) During patch embedding, the size and aspect ratio of the kernels can be adjusted based on the input. Here, `g` and `Enc` represent the patch embedding layer $g_\theta$ and the Transformer encoder $\text{Enc}_\phi(\boldsymbol{z})$.

---

[3]Bicubic resizing is implemented by `F.interpolate(x, mode="bicubic")` in PyTorch.

The implementation of multi-resolution training in Section 3.1 is as follows: We randomly select $K$ resolutions from a range of resolutions $\{r_i\}_{i=1}^{M}$, then compute the corresponding patch embeddings. After that, the model computes the loss as usual.

---

**Algorithm 1** MSPE pseudo-implementation.

---

```
1   model = ViT(...)
2   for batch in data:
3     img, label = batch
4     # create K images with random resolutions
5     hw_1, hw_2, ..., hw_K = np.random.choice([r_1, r_2, ..., r_M])
6     img_1, img_2, ..., img_K = IMG-resize(img, hw_1), IMG-resize(img, hw_2), ..., IMG-resize(img, hw_K)
7     z_1, z_2, ..., z_K = adp_conv(img1, g1, hw1), adp_conv(img2, g2, hw2), ..., adp_conv(img_K, g_K, hw_K))
8     y_1, y_2, ..., y_K = Enc(z_1), Enc(z_2), ..., Enc(z_K)
9     loss_1, loss_2, ..., loss_K = loss(y_1, label), loss(y_2, label), ..., loss(y_K, label)
10    loss = loss_1 + loss_2 ... + loss_K
11    # [...] backprop and optimize as usual
12
13  class ViT(nn.Module):
14    def __init__(self, **args, K):
15      # init as usual
16      self.N = args["img_size"]//16
17      self.g = nn.Conv2d(kernal_size=[16,16]), args)
18      self.Enc = TransformerEncoder(**args)
19      # add K patching kernels of different sizes.
20      for i in range (K):
21        sacle_i = (i + 1)/4
22        self.g_i = nn.Conv2d(kernal_size=[16 * scale_i, 16 * scale_i]), **args)
23
24    def adp_conv(self, img, func, hw ):
25      w = func.param("weight")
26      b = func.param("bias")
27      # adaptive patching kernels
28      w* = PI-resize(w, hw//self.N)
29      patch_embedding = conv(image, w*) + b
30      return patch_embedding
```

---

**Notes**: Changes to existing code highlighted via red background.

