# OpenReview forum: "MSPE: Multi-Scale Patch Embedding Prompts Vision Transformers to Any Resolution"
_NeurIPS.cc/2024/Conference — NeurIPS 2024 poster_

### Official Review · Reviewer_9L8r · 2024-07-10

**Soundness:** 3
**Presentation:** 3
**Contribution:** 3
**Rating:** 7
**Confidence:** 4

**Summary:**

1. They analyze the resolution adaptability in ViT models, identify the patch embedding layer as the crucial component, and provide a low-cost solution.

2. They propose Multi-Scale Patch Embedding (MSPE), which enhances ViT models by substituting the standard patch embedding layer with learnable, adaptive convolution kernels, enabling ViTs to be applied on any input resolution.

3. Experiments demonstrate that with minimal training (only five epochs), MSPE significantly enhances performance across different resolutions on classification, segmentation, and detection tasks.

**Strengths:**

1. They found two problems in the optimization target function in FlexiViT and they illustrate similarity in patch embeddings does not ensure the best performance.

2. MSPE allows ViTs to handle variable input resolutions directly, significantly improving performance on low-resolution inputs and maintaining high performance on high-resolution inputs.

3. MSPE is easy to apply to existing ViT models as it only modifies the patch embedding layer, avoiding the need for high-cost training or extensive model modifications.

4. MSPE significantly improves performance across different visual tasks, including image classification, segmentation, and detection, especially in comparison to previous methods like FlexiViT and ResFormer.

**Weaknesses:**

1. This method assumes that the input x follows a normal distribution x∼N(0,1) for finding the optimal solution in downscaling scenarios. However, this assumption may not hold true for all input images, particularly in real-world scenarios where image data often does not follow a normal distribution.

2. Equation (3) isolates the optimization to gθ, but a more holistic approach that includes Encϕ could potentially lead to better alignment of the entire model's parameters, ensuring more coherent and efficient learning across different resolutions.

**Questions:**

It's impressive that MSPE achieves much better results with low-resolution images. Is it possible to also show the performance with higher image resolutions larger than 1000?

---

> ### Author Rebuttal · Authors · 2024-08-06
>
> Thank you for providing us with your valuable feedback and suggestions. We appreciate your input and have carefully considered your questions. Below, we provide detailed responses to each of them:
>
> > #### **W1: Assuming $\mathcal{X} \sim \mathcal{N}(0, 1)$ in Image Processing**
>
> We completely agree with this point. FlexiViT aims to equalize patch embeddings, using the assumption $\mathcal{X} \sim \mathcal{N}(0, 1)$ for downscaling solutions.
> However, in real-world scenarios, image pixels do not necessarily meet this assumption, leading to poor FlexiViT performance at low resolutions.
>
> In light of this issue, we introduce learnable patch embedding and propose a more reasonable optimization condition that minimizes loss across different resolutions, which could compensate the limitation of the above assumption. In other words, our method does not rely on  $\mathcal{X} \sim \mathcal{N}(0, 1)$ assumption, and the experimental results in Sec. 4 demonstrate that our method significantly outperforms FlexiViT on low-resolution inputs. We would revise the motivation part (Sec. 2.4) to clarify it.
>
>
> > #### **W2: Exploring Optimization of Encoder**
>
> We appreciate the reviewer's insightful question. In this study, we aim to apply minimal modifications to various pre-trained ViT models, finding that optimizing only the patch embedding layer achieves excellent performance. Indeed, optimizing the encoder can increase the model's performance but adds computational costs. Furthermore, optimizing the encoder must be handled carefully to avoid degrading existing representations and causing knowledge forgetting.
>
>  Optimizing the patch embedding layer does not impact the original model's performance, as MSPE only optimizes added convolutional kernel parameters. Below are the experimental results for optimizing the encoder, showing performance improvements at other resolutions but a decrease at the standard 224x224 resolution. We hope these results address your concerns about encoder optimization. We will add the above discussions in the revised version.
>
>
> | **Method**   | **Resolution** |            |            |            |            |
> | :----------- | :------------: | :--------: | :--------: | :--------: | :--------: |
> |              | 56             | 112        | 168        | 224        | 448        |
> | Vanilla      | 54\.67         | 78\.75     | 83\.66     | 85\.10     | 53\.81     |
> | MSPE         | 77\.94         | 83\.75     | 84\.70     | **85\.10** | **85\.11** |
> | MPSE+Encoder | **79\.33**     | **83\.94** | **84\.75** | 84\.73     | 85\.07     |
>
>
> > #### **Q1: Testing on High-Resolution Images**
>
> Thank you for bringing up this important point. Below are our experimental results at higher resolutions, where we tested two types of ViT models, including Non-overlapping patch embedding (ViT) and overlap patch embedding (PVT). We will add the results of high-resolution in the revised version.
>
> |     | **Method** | **Resolution** |            |            |            |            |            |            |            |            |            |
> | :-: | :--------- | :------------: | :--------: | :--------: | :--------: | :--------: | :--------: | :--------: | :--------: | :--------: | :--------: |
> |     |            | 224            | 448        | 672        | 896        | 1120       | 1792       | 2240       | 2688       | 3360       | 4032       |
> | ViT | Vanilla    | 85\.10         | 53\.04     | 5\.17      | 0\.68      | 0\.33      | 0\.13      | 0\.08      | 0\.08      | 0\.08      | 0\.08      |
> |     | MSPE       | **85\.10**     | **85\.11** | **85\.13** | **85\.14** | **85\.14** | **85\.15** | **85\.14** | **85\.14** | **85\.13** | **85\.16** |
> | PVT | Vanilla    | 83\.12         | 73\.64     | 71\.18     | 67\.04     | 67\.04     | 64\.19     | 63\.73     | 63\.50     | 64\.44     | 63\.26     |
> |     | MSPE       | **83\.12**     | **82\.44** | **83\.11** | **82\.76** | **83\.02** | **82\.89** | **82\.93** | **82\.65** | **82\.97** | **82\.58** |

---

> > ### Comment · Reviewer_9L8r · 2024-08-07
> >
> > Thank you for clarifying the weaknesses. Your explanation has resolved my concerns.
> >
> > I'm glad to see the tremendous performance on the higher resolution images.
> >
> > I maintain my score as accepted.

---

### Official Review · Reviewer_gLBy · 2024-07-12

**Soundness:** 2
**Presentation:** 2
**Contribution:** 2
**Rating:** 5
**Confidence:** 4

**Summary:**

The paper introduces Multi-Scale Patch Embedding (MSPE), a novel approach to enhance Vision Transformers (ViTs) by allowing them to adapt to variable input resolutions without resizing images to a fixed resolution. MSPE uses multiple variable-sized patch kernels and selects the optimal parameters for different resolutions, maintaining performance across tasks likes image classification, segmentation, and detection.

**Strengths:**

1. MSPE replaces the standard patch embedding layer in Vision Transformers (ViTs) with multiple learnable adaptive convolution kernels, enabling the effective handling of different input resolutions.

2. MSPE demonstrates improved performance across varying resolutions in classification, segmentation, and detection tasks with minimal additional training.

**Weaknesses:**

1. This paper does not experiment with other more effective positional encoding strategies. The paper mentions that linear interpolation of positional embedding has achieved acceptable performance in existing work.

2. The paper analyses the importance of patch embedding but lacks corresponding experimental evidence comparing its importance to other components like encoder and positional encoding. Additionally, the components may show different impacts for different tasks; it does not analyze how the importance of these components may vary across different tasks.

3. The training efficiency of MSPE is highlighted, yet the potential long-term benefits of additional training epochs are not fully explored. It would be beneficial to see more detailed comparisons over extended training periods to understand better the model’s convergence behavior and potential overfitting issues.

4. Minor Error: In 2.1 Vision Transformer Models, the shape of image `x` is defined as `h*w*c`, but the kernel shape obtained is `h_k * w_k * d`. Shouldn't the kernel shape be `h_k * w_k * c * d`?

**Questions:**

1. Is selecting the hyperparameter λ during the training process crucial for the model's performance? Can different values of λ yield good results?

2. Can you elaborate on the differences between MSPE with FlexiViT?

3. Is there a rationale behind selecting the hyperparameter K and its corresponding shape in MSPE, or was it arbitrarily chosen?

4. Did the authors attempt to apply the model to uncommon resolutions (e.g., 200x150) to test its generalization performance?

---

> ### Author Rebuttal · Authors · 2024-08-06
>
> Thank you for your valuable feedback. We have carefully considered your questions and would like to address them as below:
> > #### **W1: Experiments of Positional Encoding Strategies**
>
> Thanks for your comment. MSPE employs the vanilla position encoding approach, a learnable $(N, DIM)$ vector where $N = \left\lfloor \frac{h}{h_k} \right\rfloor \times \left\lfloor \frac{w}{w_k} \right\rfloor$ For non-standard input resolutions, we resize the $(N, DIM)$ vector to $(N_{new}, DIM)$ using linear interpolation, commonly used in ViT models as **_resample absolute position embedding_**.
>
> Experiments in Sec. 4 demonstrate that, using the vanilla position encoding strategy, MSPE significantly outperforms other methods with complex position encoding strategies, such as ResFormer and NaviT.
> Actually, we intend not to use other complex positional encoding methods as they do not align with pre-trained ViT models. As shown in the results below, changing the position encoding significantly reduces the performance of pre-trained models (ViT/B-16), and retraining does not recover the expected performance.
> | **Method**               | **Training Epoch** |        |        |        |        |
> | :----------------------- | :----------------: | :----: | :----: | :----: | :----: |
> |                          | 0                  | 5      | 10     | 20     | 50     |
> | Learned 2D (NaViT)       | 0\.10              | 80\.36 | 81\.29 | 81\.85 | 82\.33 |
> | Global-Local (ResFormer) | 0\.10              | 77\.97 | 78\.80 | 79\.54 | 80\.42 |
> | Vanilla                  | **85\.10**         | -      | -      | -      | -      |
>
> > #### **W2: Analysis of Patch Embedding and Its Relative Importance**
>
> Thank you for this comment. We focus on improving patch embedding rather than the encoder or positional encoding for two reasons.
> - **First**, as demonstrated by the analysis in W1, altering the positional encoding strategy necessitates retraining the trained ViT models, incurring significant costs.
> - **Second**, modifying the encoder presents similar challenges. Our goal, as detailed in motivation (Sec. 2.4), is to adapt pre-trained ViT models to different resolutions with minimal changes.
>
> Modifying patch embedding is easy to implement, compatible with pre-trained ViT models, and significantly improves performance at various resolutions without compromising the original performance.
>
>
> - Does the analysis yield different conclusions for different tasks?
>
> In Sec. 2.4, we examined the impact of patch embedding on classification tasks by analyzing the class token at various resolutions. In classification, the class token carries global semantic information and serves as the feature for classification. In segmentation and detection tasks, such as SETR and ViTDeT, the class token is similarly utilized as the feature map for segmentation and detection. Thus, the analysis of the class token is consistent in principle across these segmentation and detection models. The experimental results in Tables 5 and 6 support our findings, demonstrating performance improvement in segmentation and detection tasks.
>
>
> > #### **W3: Analysis of Extended Training Epochs**
>
> Thanks for your suggestion. We have provided results for **1, 3, 5, 10, and 20** training epochs in Table 5 of Appendix B, which we hope will address your concerns. These results show that MSPE stabilizes after 5 epochs, and 20 epochs result in slight improvements without overfitting issues. One advantage of our method is that it requires only 5 training epochs, because MSPE only modifies the patch embedding layer of the ViT model and has few trainable parameters. Therefore, very few training epochs are needed.
>
>
> > #### **W4: Minor Error**
>
> Thank you for highlighting this error. We agree with your suggestion about the kernel shape, and we corrected it in the revised version of the paper.
>
> > #### **Q1, Q3: Analysis of $\lambda$ and $K$**
>
> - The loss function is defined as:
> $$\mathcal{L}_{\theta}(x,y)= \sum _{i=1}^K  \ell [(\text{Enc} _{\phi} (g _{\widehat{\theta} _i} (B _{r}^{r _i}x)), y ] + \lambda \cdot \ell [(\text{Enc} _{\phi}(g _{\theta}(x)), y ] $$
> $\lambda$ is a hyperparameter to prevent performance degradation. The ablation study in Figure 7(a) presents results for $\lambda$ values of 0, 1, and 2, indicating:
> **(1)** The parameter is essential for maintaining performance, with $\lambda=1$ enhancing accuracy by 4.2% over $\lambda=0$.
> **(2)** The hyperparameter shows limited sensitivity, as increasing from $\lambda=1$ to $\lambda=2$ makes little difference.
> The revised version will include a wider range of $\lambda$ results.
>
> - $K$ represents the number of convolutional kernels used for patch embedding. It is impractical to use a unique kernel size for each resolution. In MSPE, both the size and ratio of kernels $(w_{\theta}^i, b_{\theta}^i)$ are adjustable. The ablation study in Figure 7(b) demonstrates that $K=4$ is adequate, employing kernel sizes of 16x16, 12x12, 8x8, and 4x4.
>
> > #### **Q2: Differences between MSPE and FlexiViT**
>
> FlexiVit optimizes patch embeddings consistently across resolutions through explicit analytical transformations. However, it has two disadvantages:
> - **First**, it imposes a strict sufficient condition for handling varying resolutions, but the similarity in patch embeddings does not ensure optimal performance.
> - **Second**, in the downscaling scenario, there is no analytical solution for this goal.
>
> Differently, MSPE optimizes patch embeddings by minimizing the objective function across different resolutions, employing multiple dynamically adjustable patch embedding kernels to handle different resolutions effectively. Experimental results demonstrate that MSPE significantly outperforms FlexiVit.
>
> > #### **Q4: Test at Non-Standard Resolutions**
>
> In Figures 1, 4, and 9 of our paper, we tested many uncommon resolutions, such as 192x144, 80x192, 896x128, etc. Our method can also be applied to resolutions like 200x150.

---

> ### Comment · Reviewer_gLBy · 2024-08-13
>
> The response has resolved my concerns, and I have raised the score.

---

### Official Review · Reviewer_RdsV · 2024-07-13

**Soundness:** 4
**Presentation:** 4
**Contribution:** 4
**Rating:** 7
**Confidence:** 4

**Summary:**

This paper proposes to substitute the standard patch embedding with multiple variable-sized patch kernels. This eliminates the need to resize the original image. Extensive experiment results are shown to demonstrate the benefits.

**Strengths:**

The problem is well defined and the proposed method is sound. Convincing results are shown to support the claims.

**Weaknesses:**

NA

**Questions:**

The highest resolution shown is 448, 2x the training resolution 224. A typical cell phone camera has a resolution around 2K. Could the proposed method bridge this 10x gap?

**Limitations:**

yes

---

> ### Author Rebuttal · Authors · 2024-08-06
>
> Thank you for providing valuable feedback and suggestions. We greatly appreciate your input and completing the necessary experiments. Here is our detailed response to your questions:
>
> >**Q: Experimental Results on More Than 10x Resolution**
>
> Our method can be directly applied to high-resolution images because the shape and size of the patch embedding kernels dynamically adjust according to the input resolution and have multiple convolution kernels corresponding to different resolution ranges. Below are our experimental results for high-resolution images. We tested models with non-overlapping patch embedding (ViT) and overlapping patch embedding (PVT). We will include those results in the revised version.
>
> |     | **Method** | **Resolution** |            |            |            |            |            |            |            |            |            |
> | :-: | :--------- | :------------: | :--------: | :--------: | :--------: | :--------: | :--------: | :--------: | :--------: | :--------: | :--------: |
> |     |            | 224            | 448        | 672        | 896        | 1120       | 1792       | 2240       | 2688       | 3360       | 4032       |
> | ViT | Vanilla    | 85\.10         | 53\.04     | 5\.17      | 0\.68      | 0\.33      | 0\.13      | 0\.08      | 0\.08      | 0\.08      | 0\.08      |
> |     | MSPE       | **85\.10**     | **85\.11** | **85\.13** | **85\.14** | **85\.14** | **85\.15** | **85\.14** | **85\.14** | **85\.13** | **85\.16** |
> | PVT | Vanilla    | 83\.12         | 73\.64     | 71\.18     | 67\.04     | 67\.04     | 64\.19     | 63\.73     | 63\.50     | 64\.44     | 63\.26     |
> |     | MSPE       | **83\.12**     | **82\.44** | **83\.11** | **82\.76** | **83\.02** | **82\.89** | **82\.93** | **82\.65** | **82\.97** | **82\.58** |

---

> > ### Comment · Reviewer_RdsV · 2024-08-12
> >
> > Thanks for adding new results. I assume these are up sampled images and hence little change in accuracy. If so, please make it clear in the final version.

---

### Official Review · Reviewer_tsgU · 2024-07-15

**Soundness:** 3
**Presentation:** 3
**Contribution:** 2
**Rating:** 6
**Confidence:** 4

**Summary:**

The paper aim to address the challenge of adapting ViTs to variable input resolutions, which is a critical issue often overlooked in real-world applications. The authors propose a new method named Multi-Scale Patch Embedding (MSPE), which enhances the patch embedding layer by incorporating multiple variable-sized patch kernels. This approach allows the model to process images of varying resolutions without resizing, thus maintaining performance across different resolutions.

**Strengths:**

1. Problem Identification: This paper clearly identifies the gap in current ViT models, i.e. they are unable to handle variable input resolutions effectively. This is a crucial problem as real-world images often come in various resolutions.
2. Sound Approach: The proposed MSPE substitutes the standard patch embedding layer with learnable adaptive convolution kernels, which allows the model to adjust to different input resolutions dynamically.
3. Theoretical Analysis: The paper offers a thorough theoretical analysis of the problem and the proposed solution. This includes a detailed discussion on the patch embedding layer's role and the limitations of existing methods like FlexiViT and ResFormer.
4. Comprehensive Experiments: The authors provide extensive experimental results demonstrating the effectiveness of MSPE across various tasks, including image classification, segmentation, and detection.

**Weaknesses:**

1. Technical Contribution: Although the proposed method improves the multi-resolution performance of ViTs, it essentially solves an engineering problem from an engineering perspective.
2. Scalability Concerns: Although the method is low-cost and compatible with existing ViT models, the paper does not fully address potential scalability issues when applied to very large datasets or extremely high-resolution images.
3. Ablation Studies: The paper could benefit from more detailed ablation studies to isolate the contributions of different components of the MSPE and better understand the impact of each design choice.
4. Real-World Applications: While the experiments are comprehensive, the paper could include more real-world application scenarios to demonstrate the practical benefits and robustness of MSPE in diverse settings.

**Questions:**

See Weakness.

**Limitations:**

The authors addressed the limitation in the paper.

---

> ### Author Rebuttal · Authors · 2024-08-06
>
> We greatly appreciate your insightful comments and suggestions, as they have been helpful in refining and enhancing our work. We have thoroughly reviewed all of your points and have addressed your concerns as outlined below:
>
> > #### **W1: Technical Contribution**
>
> Thank you for your kind feedback. In fact, images of different sizes and aspect ratios represent different data distributions, and effectively handling these varying resolutions is critical to addressing data distribution challenges in open-world settings. This is particularly crucial for real-world applications, as image resolutions are diverse, not fixed. However, ViTs lack robustness to different resolutions. As demonstrated in our experiments (Sec. 4.1), a model used for a 224x224 resolution significantly underperforms when processing inputs at 448x448 or 128x128 resolutions.
>
> Previous works faced two major issues: **(1)** introduced complex modules and strategies that are incompatible with existing ViTs architectures, requiring the entire model to be retrained; **(2)** insufficient adaptability across various sizes and aspect ratios, limiting their performance to multiple resolutions.
> Our method does not require costly training or complex modifications, making it easily applicable to most ViT models. Moreover, it significantly outperforms previous methods. The above discussion will be added in the revised version.
>
>
>
>
> > #### **W2: Scalability Concerns**
>
> We agree that scalability is crucial for method applicability. In Sec. 4 of our paper, we conducted extensive experiments across multiple scenarios and datasets, including ImageNet-1K, COCO2017, ADE20K, and Cityscapes, as well as on various ViT models like ViT, DeiT III, and PVT.
> To further address your concerns about scalability, we also performed high-resolution tests on ImageNet-1K. The following experimental results demonstrate that our method performs excellently across large datasets and high-resolution settings. We will add those results to the revised version.
>
> |     | **Method** | **Resolution** |            |            |            |            |            |            |            |            |            |
> | :-: | :--------- | :------------: | :--------: | :--------: | :--------: | :--------: | :--------: | :--------: | :--------: | :--------: | :--------: |
> |     |            | 224            | 448        | 672        | 896        | 1120       | 1792       | 2240       | 2688       | 3360       | 4032       |
> | ViT | Vanilla    | 85\.10         | 53\.04     | 5\.17      | 0\.68      | 0\.33      | 0\.13      | 0\.08      | 0\.08      | 0\.08      | 0\.08      |
> |     | MSPE       | **85\.10**     | **85\.11** | **85\.13** | **85\.14** | **85\.14** | **85\.15** | **85\.14** | **85\.14** | **85\.13** | **85\.16** |
> | PVT | Vanilla    | 83\.12         | 73\.64     | 71\.18     | 67\.04     | 67\.04     | 64\.19     | 63\.73     | 63\.50     | 64\.44     | 63\.26     |
> |     | MSPE       | **83\.12**     | **82\.44** | **83\.11** | **82\.76** | **83\.02** | **82\.89** | **82\.93** | **82\.65** | **82\.97** | **82\.58** |
>
>
>
> > #### **W3: Ablation Studies**
>
> Thank you for your suggestions. We conducted extensive ablation experiments in Sec. 4.4 and Appendix D, including training epochs, model size, hyperparameters, kernel count (K), and resizing methods. The results are displayed in Figures 6, 7, 8, 10, and Table 6.
> The experiments show that our method is robust to hyperparameters and that increasing training epochs and kernel counts boosts performance. We achieve good performance and basic convergence with 5 training epochs and 4 kernels.
> We also evaluated different resizing methods, including nearest, bilinear, and bicubic, with PI-resize proving the most effective, as shown in Figures 6 and 8. We hope these detailed results address your concerns about ablation studies. We would revise the ablation study part (Sec. 4.4) to make it clearer.
>
>
>
>
> > #### **W4: Real-World Applications**
>
> We completely agree with this. Adapting to different resolutions is a key challenge for ViT models in real-world applications. Our method was evaluated on multiple real-world datasets such as Cityscapes, ADE20K, ImageNet-1K, and COCO2017, including tasks like classification, detection, and segmentation.
> The City dataset collects street scenes from various cities; ADE20K includes scenes from more than 150 settings; ImageNet-1K and COCO2017 are extensive, covering numerous real-world environments.
>
> Tables 5 and 6 in our paper show that MSPE significantly improves across various resolutions and task settings. We hope these experiment results confirm the substantial potential for real-world application.

---

> > ### Comment · Reviewer_tsgU · 2024-08-13
> >
> > The rebuttal addresses most of my concerns. I would raise my score.

---

### Author Rebuttal · Authors · 2024-08-07

We sincerely thank the reviewers for their detailed and perceptive comments, which have significantly refined and improved this paper.

We are grateful that the reviewers appreciate our paper in various aspects, including its well-defined problem and theoretical analysis [tsgU, RdsV, 9L8r], simple but solid method [tsgU, RdsV, gLBy, 9L8r], and remarkable performance [tsgU, RdsV, gLBy, 9L8r].

In this work, we explore the challenges of applying ViT models at different resolutions and propose MSPE, which improves ViT models by replacing the standard patch embedding layer with multiple learnable, adaptive convolution kernels. This makes ViTs adaptable to any input resolution without requiring high-cost training or extensive model modifications.


***
Reviewers are particularly interested in how our method performs at high resolutions. Below are the results at high resolution. The results demonstrate that MSPE significantly enhances performance at high resolutions, enabling a single ViT model to effectively cover resolutions from 28x28 to 4032x4032, which previous methods could not achieve.

|     | **Method** | **Resolution** |            |            |            |            |            |            |            |            |            |
| :-: | :--------- | :------------: | :--------: | :--------: | :--------: | :--------: | :--------: | :--------: | :--------: | :--------: | :--------: |
|     |            | 224            | 448        | 672        | 896        | 1120       | 1792       | 2240       | 2688       | 3360       | 4032       |
| ViT | Vanilla    | 85\.10         | 53\.04     | 5\.17      | 0\.68      | 0\.33      | 0\.13      | 0\.08      | 0\.08      | 0\.08      | 0\.08      |
|     | MSPE       | **85\.10**     | **85\.11** | **85\.13** | **85\.14** | **85\.14** | **85\.15** | **85\.14** | **85\.14** | **85\.13** | **85\.16** |
| PVT | Vanilla    | 83\.12         | 73\.64     | 71\.18     | 67\.04     | 67\.04     | 64\.19     | 63\.73     | 63\.50     | 64\.44     | 63\.26     |
|     | MSPE       | **83\.12**     | **82\.44** | **83\.11** | **82\.76** | **83\.02** | **82\.89** | **82\.93** | **82\.65** | **82\.97** | **82\.58** |



***
For other specific questions, we provide detailed responses to each reviewer below. We will carefully revise the paper based on all the feedback from the four reviewers. Thank you once again for your valuable suggestions!

---

### Comment · Area_Chair_bqCf · 2024-08-12
**Reviewer-author discussion period**

Dear reviewers,

The authors have responded according to your comments. Please have a check to see whether your concerns are adequately solved. You can ask authors to address further if required.

Best,
Your AC

---

### Decision · Program_Chairs · 2024-09-25

**Decision:**

Accept (poster)

**Comment:**

The manuscript received all positive ratings. The reviewers appreciated the soundness of the proposed approach, theoretical analysis and comprehensive experiments. Reviewers also raised several issues which were addressed in the rebuttal. Given the reviewers comments, rebuttal and discussion, the recommendation is accept. Authors are encouraged to take into account all the reviewers feedback and rebuttal content when preparing the final version.